# Perturbations in Traffic: Aberrant Nucleocytoplasmic Transport at the Heart of Neurodegeneration

**DOI:** 10.3390/cells7120232

**Published:** 2018-11-26

**Authors:** Birthe Fahrenkrog, Amnon Harel

**Affiliations:** 1Institute of Molecular Biology and Medicine, Université Libre de Bruxelles, 6041 Charleroi, Belgium; 2Azrieli Faculty of Medicine, Bar-Ilan University, Safed 1311502, Israel; amnon.harel@biu.ac.il

**Keywords:** nucleocytoplasmic transport, importin, transportin, neurodegenerative disease, ALS, FTD, Huntington’s disease

## Abstract

Neurodegenerative diseases, such as amyotrophic lateral sclerosis (ALS), frontotemporal dementia (FTD), and Huntington’s disease (HD), are characterized by intracellular aggregation of proteins. In the case of ALS and FTD, these protein aggregates are found in the cytoplasm of affected neurons and contain certain RNA-binding proteins (RBPs), namely the TAR DNA-binding protein of 43 kDa (TDP-43) and the fused in sarcoma (FUS) gene product. TDP-43 and FUS are nuclear proteins and their displacement to the cytoplasm is thought to be adverse in at least two ways: loss-of-function in the nucleus and gain-of-toxicity in the cytoplasm. In the case of HD, expansion of a polyglutamine (polyQ) stretch within the N-terminal domain of the Huntingtin (HTT) protein leads to nuclear accumulation of polyQ HTT (or mHTT) and a toxic gain-of-function phenotype resulting in neurodegeneration. Numerous studies in recent years have provided evidence that defects in nucleocytoplasmic transport critically contribute to the pathology of these neurodegenerative diseases. A new mechanistic view is emerging, implicating three types of perturbations in normal cellular pathways that rely on nucleocytoplasmic transport: displacement of nuclear transport receptors and nucleoporins from nuclear pore complexes (NPCs), mislocalization and aggregation of RNA-binding proteins, and weakening of the chaperone activity of nuclear import receptors.

## 1. Defects Point to Nuclear Pores

Neurodegenerative diseases are associated with the aggregation of misfolded proteins, which are toxic to the affected neurons and eventually cause neuronal cell death. In the case of amyotrophic lateral sclerosis (ALS) and frontotemporal dementia (FTD), aggregates of two RNA-binding proteins (RBPs), the TAR DNA-binding protein of 43 kDa (TDP-43) and the fused in sarcoma (FUS) gene product, are found in the cytoplasm of affected neurons [1,2]. By contrast, Huntington’s disease (HD) presents with the nuclear accumulation of mutant Huntingtin (mHTT), which arises from the expansion of a polyglutamine (polyQ) stretch within the N-terminal domain of the HTT protein [3]. Recent work implicates disrupted nuclear pore complexes (NPCs) and dysregulated nucleocytoplasmic transport as a common disease mechanism in ALS, FTD and HD.

Nuclear pore complexes (NPCs) span the double membranes of the nuclear envelope (NE) and function as the sole transport conduit between the nucleus and the cytoplasm of eukaryotic cells. NPCs consist of distinct structural subunits, namely the central scaffold, the cytoplasmic and nuclear rings, the cytoplasmic filaments, the nuclear basket, and the central pore channel (Figure 1A) [4,5]. On the molecular level, NPCs are composed of roughly 30 different nucleoporins (Nups), out of which about one third contain repetitive phenylalanine-glycine (FG) motifs. These FG motifs in nucleoporins mediate the critical interactions with soluble nuclear import and export receptors, known as karyopherins or importins and exportins, that carry molecular cargoes through the NPC channel [6]. The directionality of nucleocytoplasmic transport is determined by the small GTPase Ran and its regulators RCC1 (the Ran guanine nucleotide exchange factor) and RanGAP1 (the Ran GTPase activating protein). RCC1 binds to chromatin, whereas RanGAP1 localizes exclusively to the cytoplasmic filaments of the NPC (Figure 1B). This compartmentalization of the Ran regulators facilitates the establishment of a Ran gradient within the cell, with high RanGTP levels inside the nucleus and high RanGDP concentration in the cytoplasm [6].

Defects in NPCs were first described in the context of HD. The HTT protein localizes to both the cytoplasm and the nucleus and contains a so-called PY-NLS, a nuclear localization signal enriched in proline-tyrosine residues, as well as a nuclear export signal (NES; Figure 2) [7,8,9]. Its nuclear import is mediated by two import receptors: importin-β and transportin-1 (also known as importin-β2) [8]. In the pathological condition of HD, mHTT aggregates in intranuclear inclusions within cells in the striatum and the cortex regions of the brain. These inclusions contain, amongst others, NUP62, a nucleoporin which is normally situated around the central pore channel of the NPC (Figure 1B) [10]. Furthermore, a quantitative interaction proteomics screen revealed that RanGAP1, the mRNA export factor Rae1 (ribonucleic acid export 1), and the nucleoporin Sec13 (Figure 1B) preferentially interact with mHTT, whereas the nuclear import receptors importin-β1, importin-4, importin-7, and importin-9 showed preferred binding to wild-type HTT [11]. The association of the FG-repeat nucleoporin NUP62 and RanGAP1 with mHTT intranuclear inclusions was more recently confirmed in a HD mouse model [12]. These intracellular inclusions additionally contained NUP88 (Figure 1B) [12], which is a component of the cytoplasmic ring [13] but is also located on the nuclear side of the NPC [14], and the mRNA export factor Gle1 [15]. In post-mortem brain tissue from HD and juvenile HD (JHD) patients, NUP62 did not aggregate, but mislocalized to both the nucleus and the cytoplasm [12], while RanGAP1 was sequestered into mHTT nuclear aggregates [15]. 

NUP62, RanGAP1 and Ran are consistently mislocalized in HD pathology. This was observed in cultured neurons differentiated from induced pluripotent stem (iPSN) cells derived from HD patients, as well as in mHTT transfected mouse neurons [12]. By contrast, simultaneous overexpression of mHTT alongside with RanGAP1-GFP or Ran-GFP was neuroprotective, both in mouse neurons and in a *Drosophila* HD model [12]. Neuroprotection was manifested in reduced cell death and increased viability of the transfected neurons [12]. 

NUP62 and other nucleoporins are modified by *O*-linked *N*-acetylglucosamine (*O*-GlcNAc), although the functional role of this posttranslational modification is largely unknown [16,17]. Surprisingly, the treatment of primary mouse mHTT-expressing cortical neurons with an inhibitor of *O*-GlcNAc transferase rescued the nucleocytoplasmic transport defects and restored the proper localization of Ran and an import–export cargo reporter. Similarly, specific inhibitors of the nuclear export receptor CRM1 reversed nucleocytoplasmic transport defects, suggesting that the inhibition of nuclear export may be neuroprotective by compensating nuclear import defects under the pathological conditions of HD [12].

Like the situation in HD, nucleoporins and RanGAP1 are also displaced from NPCs in ALS pathology. One of the best studied genetic causes for ALS and FTD is an abnormal hexanucleotide GGGGCC (G_4_C_2_) repeat expansion in the *C9orf72* gene [18,19]. RanGAP1 forms accumulations that co-localize with cytoplasmic RNA foci in C9orf72 ALS (C9-ALS) postmortem brain tissue, in iPSN cells derived from C9-ALS patients, and in *Drosophila* cells expressing such G_4_C_2_ repeats [20]. Several nucleoporins, including the scaffold components NUP205 and NUP107 (Figure 1B), also accumulate in C9-ALS patient brain tissue and iPSNs. A similar cytoplasmic aggregation of nucleoporins and RanGAP1 was observed in SOD1-ALS [21,22]. SOD1-ALS is caused by mutations in the human Cu/Zn superoxide dismutase 1 (SOD1) and is causative for about 23% of familial ALS and 3% of sporadic ALS [23,24]. These cytoplasmic aggregates were recently identified as cytoplasmic stress granules (SGs) [25]. SGs are dynamic cytoplasmic RNA–protein complexes that contain RBPs, mRNAs, and translation initiation factors, and appear to play an important role in ALS and FTD pathology [26]. SGs also contain a distinct set of importins and exportins as well as Ran, but lack both Ran regulators, RCC1 and RanGAP1.

The association of nucleoporins and transport factors with SGs was confirmed in various experimental models including C9-ALS iPSNs and C9-ALS *Drosophila* models. A proximity based biotin identification (BioID) assay and a genetic screen in *Drosophila* revealed nucleoporins and nuclear export factors as genetic modifiers and components of SGs in C9-ALS [27,28]. Fifteen individual nucleoporins, the exportin Xpo5 and the mRNA export factors NFX1 and Gle1 were shown to co-aggregate with TDP-43 in these C9-ALS *Drosophila* models [27]. Loss-of-function mutations in several *Drosophila* nucleoporin genes rescued the phenotypes caused by TDP-43 pathology, whereas other nucleoporins acted as enhancers [27,28]. Importantly, some aspects of neurodegeneration could be reversed by either the inhibition of SG assembly, achieved by silencing of CRM1, or by the inhibition of CRM1-mediated nuclear export [25,27,29].

## 2. Nucleocytoplasmic Transport of RNA-Binding Proteins

As outlined above, ALS and FTD are characterized by mislocalization and aggregation of RBPs, in particular TDP-43 and FUS in the cytoplasm of degenerating neurons. What leads to the mislocalization of these predominantly nuclear proteins? Impaired nuclear import is suggested to be a key initiation event in ALS/FTD pathogenesis and a pathogenic mechanism [30,31,32]. FUS harbors a PY-NLS (Figure 3) and the majority of pathogenic FUS mutations affect this NLS [30]. Nuclear import of FUS is dependent on transportin-1, through its recognition of the PY-NLS. siRNA-mediated depletion of transportin-1 or treatment of cells with specific transportin-1 inhibitors leads to a redistribution of FUS to the cytoplasm and its recruitment into SGs, in both neuronal and transfected human cell lines [33]. In addition, methylation of the FUS PY-NLS perturbs the targeting of the protein into the nucleus [34].

TDP-43 harbors a classical, arginine-lysine rich NLS (cNLS; Figure 3) and its import is mediated by the heterodimeric importin-α1/β1 [35,36]. No cNLS mutants for TDP-43 have been described in ALS or FTLD patients [30], but its removal from TDP-43 leads to the spontaneous appearance of SGs [37]. Similarly, the overexpression of FUS PY-NLS mutants results in SG formation [33]. Conversely, in yeast, overexpression of nuclear import receptors was found to suppress the toxicity of C9orf72-associated repeat expansion [38].

Nuclear import may also be a key player in the pathogenesis of polyQ diseases, which are characterized by the nuclear accumulation of expanded-repeat proteins, similarly to HD. Spinocerebellar ataxia type 3 (SCA3) is caused by a CAG expansion in the *ATXN3* gene, which leads to a polyQ expansion in the ataxin-3 protein. Ataxin-3 harbors two nuclear export signals (NESs) and a cNLS (Figure 2) and its nuclear import is mediated by importin-α3 and importin-13 [39,40]. In *Drosophila* and mouse SCA3 models, depletion of importin-α3 prevented nuclear aggregation of mutant ataxin-3 and rescued neurodegeneration, whereas overexpression of importin-α3 or CRM1 did not significantly aggravate the disease phenotype [40]. These studies show that nuclear import plays a causative role in the pathogenesis of ALS/FTD and polyQ diseases, leading to the question whether nuclear export may play a similar role. As outlined above, inhibition of SG assembly or inhibition of CRM1-mediated nuclear export were beneficial in C9-ALS *Drosophila* models [25,27,29]. Ederle et al. have recently shown that the predicted NESs in TDP-43 and FUS are not functional and that their nuclear export is CRM1-independent [41]. Moreover, nuclear export is independent of RNA-binding and of the mRNA export machinery. Ederle et al. suggested that TDP-43 and FUS leave the nucleus by passive diffusion and that retention to newly synthesized RNA sequesters them inside the nucleus and limits diffusion into the cytoplasm.

## 3. Chaperone Activity of Nuclear Import Receptors

SGs are dense, membrane-less cytosolic aggregates of proteins and RNAs that occur upon cellular stress, such as impaired translation initiation. SGs are formed by a process called liquid–liquid phase separation, which, in physics, is defined as the separation of two liquids into two phases that cannot mix. In cells, liquid–liquid phase separation occurs when a homogenous, highly concentrated mixture of molecules, such as proteins and RNAs, spontaneously separates into two co-existing phases. These two phases comprise a so-called dense phase, which is enriched for these molecules, and a dilute phase that is depleted [42,43,44]. Phase separation is known to play a role, for example, in the formation of membrane-less organelles, such as the nucleolus and other nuclear bodies, as well as P-bodies and SGs in the cytoplasm [42,45]. Phase separation of proteins is often encoded by intrinsically disordered regions within their amino acid sequence. TDP-43, FUS and other hnRNPs belong to this family of intrinsically disordered proteins (IDPs) [42,46]. IDPs often contain regions of low-complexity, which are characterized by an enrichment of disorder-promoting amino acids, such as Ser, Gly, and Gln, and by repetitive sequence motifs [42,46]. These low-complexity domains are referred to as prion-like domains (PrDLs; Figure 3), which allow IDPs to adopt multiple conformations and exchange conformational states rapidly [42,47]. Depending on specific environmental conditions, such as protein and salt concentration, and temperature, IDPs can reversibly shift from a soluble state to a gel-like, liquid droplet state. Liquid droplets may eventually mature into fibrillar solid aggregates, a typical hallmark of neurodegenerative disorders [42].

TDP-43 and FUS contain PrDLs in their C-terminal and N-terminal regions, respectively (Figure 3). Mutations that have been identified in patients with ALS and FTD cluster in these PrDLs [47]. These mutations do not affect the soluble state of the proteins. However, they promote conformational changes and β-sheet aggregation which in turn enhance liquid droplet formation and their eventual transition into solid aggregates [42,46]. Four back-to-back articles published in Cell in April 2018 show that nuclear import receptors do not merely transport hnRNPs, but can also fine-tune their aggregation tendencies due to a chaperone-like activity [48,49,50,51]. This dual property of importins was first demonstrated in 2002 [52], but has only now attracted reasonable attention, in particular in the context of neurodegenerative diseases.

As outlined above, FUS harbors a PY-NLS, which is recognized by transportin-1 and has a suspected role in the pathogenesis of ALS and FTD. In vitro fibrillization and turbidity assays have been used to assess the phase separation potential in wild type and mutant protein forms. Recombinant, purified FUS, disease-related mutant FUS, and other ALS/FTD-related hnRNPs harboring a PY-NLS formed fibrils, but no droplets, while fibrillization was inhibited by equimolar amounts of recombinant transportin-1 [48,49,50,51]. Mutant forms of transportin-1 or other karyopherins did not prevent fibrillization [50,51]. The chaperone-like activity of transportin-1 appears to be restricted to the cytoplasm, since the deletion of the PY-NLS and the addition of RanGTP diminished the inhibitory effect of transportin-1, while simultaneous overexpression of FUS reporter constructs and transportin-1 in human cells reduced FUS localization in SGs [48,51]. Transportin-1 had no spontaneous effect on the fibrillization of wild-type and mutant TDP-43, which harbors a cNLS, whereas importin-β inhibited TDP-43 fibrillization [51]. Replacing the PY-NLS with a cNLS changed the sensitivity of FUS fibrils from transportin-1 to importin-β [50]. Thus, the anti-aggregation chaperone activity of these importins is “case-sensitive” and mediated through their recognition of specific NLSs.

A striking aspect of ALS and FTD is the regional spreading of neurodegeneration between neighboring cells and local areas of the brain [51,53,54]. Spreading of neurodegeneration appears to be due to self-propagation of seeded FUS or TDP-43 aggregates. Self-propagation has been shown for both RBPs in cell culture and other in vitro models [55,56,57,58] and for TDP-43 in vivo, in mouse and zebrafish models [59,60]. The chaperone activity of nuclear import receptors may prevent such self-propagation of RBP aggregates. Transportin-1 not only inhibited seeded fibrillization of FUS, but also disaggregated existing FUS fibrils [51]. Disaggregation assays further revealed that each transportin-1 molecule extracts one FUS monomer from a FUS fibril to form a soluble transportin-1-FUS complex, which can be transported to the nucleus, so that disaggregated FUS cannot re-aggregate in the cytoplasm [51]. On an organismal level, in *Drosophila* models for ALS, depletion of transportin-1 enhanced neurodegeneration and decreased the lifespan. By contrast, transportin-1 overexpression reduced neurodegeneration and increased the lifespan [51]. It remains to be investigated if the overexpression of importin-β will have similar effects on TDP-43 aggregates and neurodegeneration.

The chaperoning activity of transportin-1 on PY-NLS bearing hnRNPs depends on multiple interactions between both components. NMR studies revealed that transportin-1 forms high-affinity interactions with the PY-NLS of FUS, but also weak interactions in particular with the PrDL region and the C-terminal arginine-glycine-glycine (RGG) motifs (Figure 3; RGG2 and RGG3) [49,50]. The PrDL and RGGs regions of FUS are the main determinants of FUS phase separation and transportin-1 disrupts phase separation by interacting with these domains [48,50]. The anti-aggregation activity of the import receptor can also be accomplished by preventing arginine-methylation in the RGG3 motif: FUS mutants lacking the RGG3-PY motif or containing arginine-to-lysine mutations (KGG3-PY) failed to phase separate and aggregate [48].

## 4. Conclusions

In summary, perturbations in normal cellular pathways that rely on nucleocytoplasmic transport are increasingly recognized as key initiation and propagation factors in the pathogenesis of neurodegenerative diseases. Displacement of nucleoporins from NPCs and nuclear transport receptors has been described in the context of ALS and FTD, as well as HD. The mechanistic consequence of such displacement remains to be elucidated, but the concept may be a general hallmark in neurodegenerative disease and was only recently also described for Alzheimer’s disease [61].

Nuclear import defects appear to be key initiating events in ALS and FTD, which lead to the mislocalization of TDP-43 and FUS to the cytoplasm and their accumulation in SGs (Figure 4). This cytoplasmic accumulation of TDP-43 and FUS may be further potentiated by defects at the level of NPCs. Changes at the NPC level may also lead to leaky nuclear pores [62] and reduced availability of nuclear import receptors, establishing a vicious cycle. Inhibition of nuclear export may compensate nuclear import defects to a certain extent, but export currently appears to play a more minor role in ALS and FTD pathogenesis as compared to import. The role of nuclear import in polyQ diseases is less well studied, but recent data in yeast models for HD have shown that defects in the ribosomal quality control machinery lead to the nuclear accumulation of mHTT and that inhibition of nuclear import rescues the toxicity of the mHTT in the yeast model [63].

Finally, the dual function of importins as transport receptors and chaperones appears to be of outmost importance to prevent phase transition and deleterious fibrillization of disease-linked RBPs, such as TDP-43 and FUS. Most importantly, specific importins are capable of reversing aberrant phase transition and re-establishing functional phase separation by extracting the aggregated RBPs from fibers and granules. Although there is still much to be learned about the new unexpected cytoplasmic functions of importins, these intriguing multipurpose proteins may become new therapeutic targets, not only for antiviral or cancer therapy [64,65], but also for the treatment of neurodegenerative diseases.

## Figures and Tables

**Figure 1 cells-07-00232-f001:**
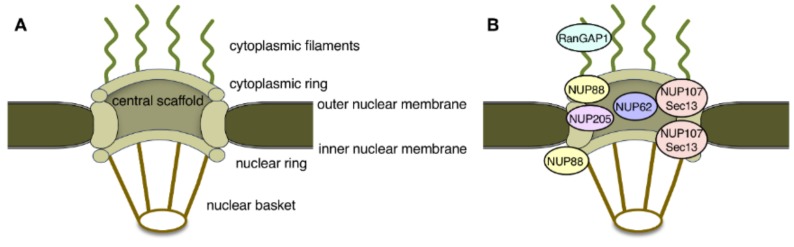
(**A**) Schematic representation of the main structural features of nuclear pore complexes (NPCs). (**B**) Localization of nucleoporins and nuclear pore associated factors implicated in neurodegenerative diseases within the 3D architecture of the NPC.

**Figure 2 cells-07-00232-f002:**
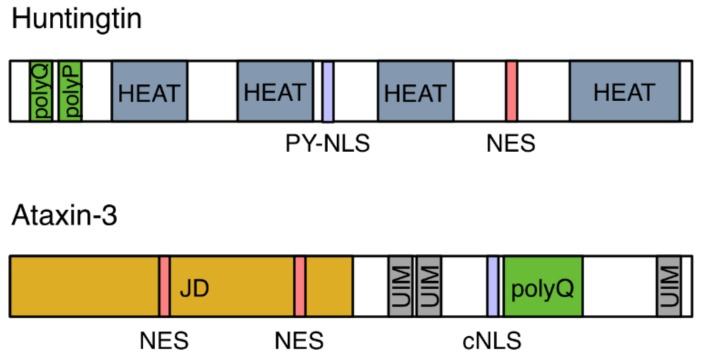
Schematic diagram of the domain architecture of Huntingtin (HTT) and ataxin-3. HTT contains polyglutamine (polyQ) and proline (polyP) repeats, a proline-tyrosine-rich NLS (PY-NLS), a nuclear export signal (NES) and four helical regions known as HEAT repeats. Ataxin-3 is composed of a globular Josephin domain (JD), three ubiquitin interacting motifs (UIM), and polyQ repeats. It harbors a classical nuclear localization signal (cNLS) and two NESs.

**Figure 3 cells-07-00232-f003:**
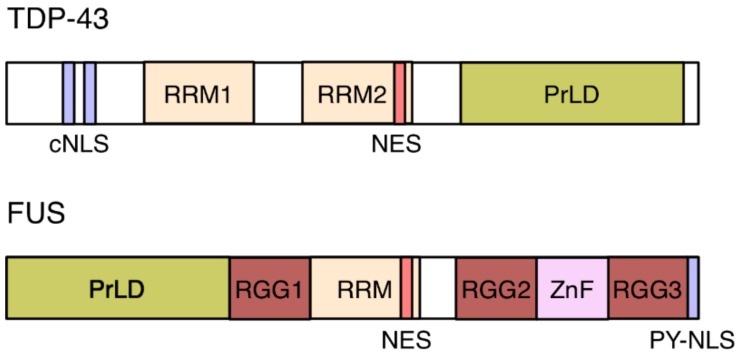
Domain structure of TDP-43 and FUS. TDP-43 contains two RNA recognition motifs (RRM), two predicted adjacent nuclear export signals (NESs), and a classical bipartite nuclear localization signal (cNLS). Its C-terminal domain harbors a low-complexity, prion-like domain (PrLD). The PrDL of FUS is located in its N-terminal half. FUS further contains two predicted adjacent NESs, one RRM, one zinc-finger (ZnF), and three arginine-glycine-glycine (RGG) repeat domains that stabilize RNA binding. FUS’s proline-tyrosine-rich NLS (PY-NLS) is located at the C terminus of the protein.

**Figure 4 cells-07-00232-f004:**
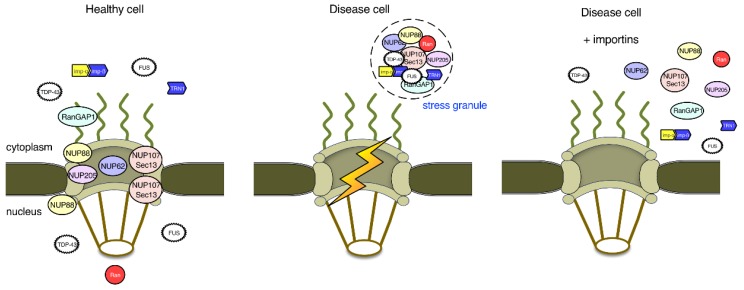
Nuclear pore proteins and nuclear import in ALS and FTD. In healthy cells, the RNA-binding proteins TDP-43 and FUS are localized to both the nucleus and the cytoplasm. In disease cells, nuclear import of TDP-43 and FUS is impaired, which leads to their aggregation into stress granules (SGs) in the cytoplasm. TDP-43 and FUS in SGs sequester several nucleoporins and factors important for the regulation of nucleocytoplasmic transport, establishing a vicious cycle. Although overexpression of importins, such as importin-β and transportin-1, may cause the dissolution of TDP-43 and FUS aggregates, whether the nucleocytoplasmic transport machinery is restored to normal function remains to be investigated.

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
