# Peer review of "Perturbations in Traffic: Aberrant Nucleocytoplasmic Transport at the Heart of Neurodegeneration"

_cells, 2018, doi:10.3390/cells7120232_

Reviewer 1 Report

Fahrenkrog and Harel summarised in this commentary three mechanisms on how disease perturbations can influence the shuttling of cargo in and out of the nucleus. This brief review highlights most of the key literature in this space and provides a good introduction into the important machineries that maintain nucleo-cytoplasmic transport in some forms of ALS and HD.

Overall the mini-review is well balanced and clearly outlines some of the (currently) key proteins affected by disease. As a commentary I would have liked to see a bit more of a personal opinion, such as examples on where the authors see the field going and where there is currently a lack of knowledge.

The two graphics could be substantially enhanced as they do not carry much beneficial information in their current form. A graphic of the specific NUPs and their localisation as well as correlation with ALS &HD would be a valuable contribution.

Specific (minor) comments:

74 The best studied genetic cause for ALS and FTD is an abnormal hexanucleotide GGGGCC

-- This is not correct as the best studies model is probably SOD1 with C9ORF72 accounting for the most cases.

84 factors, and are now an accepted hallmark of ALS and FTD [22].

-- While I think that they play an important role they are not a specific hallmark of the disease.

Lines 202-205 in the conclusion do not add much information at this part of the mini-review.

Lines 86-94: C9ORF pathology and TDP-43 pathology are written as if they were the same thing

Line 176 A striking aspect of ALS and FTD is that they spread between neighboring cells and brain regions [44,46].

-- This sentence seems to be logically detached while being an important aspect of many neurodegenerative disease. If extended, studies such as recent in vivo evidence of propagation should then be included (https://www.nature.com/articles/s41467-018-06548-9 & https://link.springer.com/article/10.1007%2Fs00401-018-1875-2)

Further, the author might consider including the following paper in their mechanistic views and references:

https://doi.org/10.1093/hmg/ddy303

Lines 204-205: ‘as outlined above’ is not a good conclusion phrase

Author Response

1. 74 The best studied genetic cause for ALS and FTD is an abnormal hexanucleotide GGGGCC

-- This is not correct as the best studies model is probably SOD1 with C9ORF72 accounting for the most cases.

We have rephrased the sentence to One of the best studied genetic cause….”(line 140).

2. 84 factors, and are now an accepted hallmark of ALS and FTD [22].

-- While I think that they play an important role they are not a specific hallmark of the disease.

We have rephrased as follows: “ factors, and appear to play an important role in ALS and FTD pathology [26].”(line 150).

3. Lines 202-205 in the conclusion do not add much information at this part of the mini-review.

We agree that the lines 202-205 (now lines 313-316) do not add much information, but act as transition for the remaining part of the paragraph. We therefore prefer to not omit these lines.

4. Lines 86-94: C9ORF pathology and TDP-43 pathology are written as if they were the same thing

 We agree with the reviewer that the text was not exactly clear. We have rephrased it therefore, so that C9ORF and TDP-43 pathology are now better distinguished (lines 153-162).

5. Line 176 A striking aspect of ALS and FTD is that they spread between neighboring cells and brain regions [44,46].

-- This sentence seems to be logically detached while being an important aspect of many neurodegenerative disease. If extended, studies such as recent in vivo evidence of propagation should then be included (https://www.nature.com/articles/s41467-018-06548-9 & https://link.springer.com/article/10.1007%2Fs00401-018-1875-2)

As suggested by the reviewer, we have extended this part a bit and added the suggested and other additional references (lines 274-286).

6. Further, the author might consider including the following paper in their mechanistic views and references:

https://doi.org/10.1093/hmg/ddy303

We have included the reference as suggested (reference 29, line 211). 

7. Lines 204-205: ‘as outlined above’ is not a good conclusion phrase

We agree with the reviewer and have deleted the phrase (lines 315-316). 

Reviewer 2 Report

This is a well-written and informative commentary regarding the relationship of nucleocytoplasmic transport defects and the pathology of neurodegenerative diseases. I only have some minor suggestions.

1.     It is not clear that where does the transmission electron micrograph come from. It might be good to clarify this by adding a reference. The micrograph seems redundant and does not provide more information than the structure figure.

2.     Instead of only showing the classical structure of nuclear pore complex, it would be more informative if the authors could expend this figure by adding the key components that were discussed in this section and how they interact with disease proteins.  It would be helpful for the readers if the author could make this figure into a graphical summary of this section.

3.     The font size in Figure 1 is a bit small after printed out, suggest to adjust to the same size as in Figure 2.

4.     For the text “central scaffold” in Figure 1, the “d” is partially covered.

5.     Figure2, since this commentary focused on the nucleocytoplasmic transport aspect, it may be better to also show the nuclear export signals of TDP-43 and FUS.

6.     The structure of ATXN3 could be added into Figure 2, since the authors spent a paragraph on discussing this gene.

7.     Figure 2 could be improved by adding the co-players of the nuclear localization signals.

8.     Relevant results using yeast model system (for example, PMID: 28864412…) could also be discussed. 

9.     The section of  “chaperon activity of unclear import receptors” discussed 4 recently published exciting reports; a schematic figure of this section will be very helpful for the readers (if possible).

Author Response

1.     It is not clear that where does the transmission electron micrograph come from. It might be good to clarify this by adding a reference. The micrograph seems redundant and does not provide more information than the structure figure.

2.     Instead of only showing the classical structure of nuclear pore complex, it would be more informative if the authors could expend this figure by adding the key components that were discussed in this section and how they interact with disease proteins.  It would be helpful for the readers if the author could make this figure into a graphical summary of this section.

We have modified Figure 1 as suggested by the reviewer. We have removed the electron micrograph and inserted a second schematic presentation of a NPC in which the discussed  nucleoporins and NPC-associated factors are highlighted.

3.     The font size in Figure 1 is a bit small after printed out, suggest to adjust to the same size as in Figure 2.

We have increased the font size in Figure 1.

4.     For the text “central scaffold” in Figure 1, the “d” is partially covered.

We moved the text so that it is no longer partially covered.

5.     Figure2, since this commentary focused on the nucleocytoplasmic transport aspect, it may be better to also show the nuclear export signals of TDP-43 and FUS.

6.     The structure of ATXN3 could be added into Figure 2, since the authors spent a paragraph on discussing this gene.

We have included the nuclear export signals of TDP-43 and FUS (now Figure 3) and a new figure showing the domain structure of Huntingtin and ataxin-3 (new Figure 2). 

7.     Figure 2 could be improved by adding the co-players of the nuclear localization signals.

In our opinion, the addition of the “co-players of the nuclear localization signals” (we assume that here the respective nuclear import factors are meant) would make the figure more confusing and would therefore prefer to leave the figure (now Figure 3) as it is.

8.     Relevant results using yeast model system (for example, PMID: 28864412…) could also be discussed. 

As suggested by the reviewer, we now refer to results obtained in yeast model systems (reference 63, line 322). 

9.     The section of  “chaperon activity of unclear import receptors” discussed 4 recently published exciting reports; a schematic figure of this section will be very helpful for the readers (if possible).

We have now included a schematic summary (new Figure 4), as suggested by the reviewer.